# Solution-Grown MAPbBr_3_ Single Crystals for Self-Powered Detection of X-rays with High Energies above One Megaelectron Volt

**DOI:** 10.3390/nano13152157

**Published:** 2023-07-25

**Authors:** Beomjun Park, Juyoung Ko, Jangwon Byun, Sandeep Pandey, Byungdo Park, Jeongho Kim, Man-Jong Lee

**Affiliations:** 1Department of Chemistry, Konkuk University, Seoul 05029, Republic of Korea; 2Advanced Crystal Material/Device Research Center, Konkuk University, Seoul 05029, Republic of Korea; 3Department of Radiation Oncology, Samsung Changwon Hospital, Sungkyunkwan University School of Medicine, Changwon 51353, Republic of Korea

**Keywords:** MAPbBr_3_, mechanochemical surface treatment, self-powered X-ray detector, perovskite single crystal, sensitivity

## Abstract

Perovskite single crystals are actively studied as X-ray detection materials with enhanced sensitivity. Moreover, the feasibility of using perovskites for self-powered devices such as photodetectors, UV detectors, and X-ray detectors can significantly expand their application range. In this work, the charge carrier transport and photocurrent properties of MAPbBr_3_ single crystals (MSCs) are improved by the mechanochemical surface treatment using glycerin combined with an additional electrode design that forms an ohmic contact. The sensitivity of MSC-based detectors and pulse shape generated by X-rays are enhanced at various bias voltages. The synthesized MSC detectors generate direction-dependent photocurrents, which indicate the presence of a polarization-induced internal electric field. In addition, photocurrent signals are produced by X-rays with energies greater than 1 MeV under a zero-bias voltage. This work demonstrates a high application potential of perovskites as self-powered detectors for X-rays with energies exceeding 1 MeV.

## 1. Introduction

The detection of X/γ-rays is widely used in various applications, such as medical diagnostics, material characterization, radiotherapy, astronomy, and nuclear power plants [1,2,3,4]. Among the utilized X/γ-ray detectors, semiconductor detectors directly convert radiation to electric signals, while scintillator detectors with indirect conversion require additional components for X/γ-rays detection and exhibit low energy resolution [5,6]. Subsequently, semiconductor radiation detectors are characterized by high sensitivity and a low minimum detectable dose as compared with those of other detector types. Moreover, direct conversion perovskite X/γ-ray detectors have been actively investigated since the pioneering work on perovskites [7,8]. These detectors possess several advantages, including a facile growth procedure, low precursor cost, low defect density, high sensitivity, and stoichiometric tunability [9,10,11,12,13].

Sensitivity is an important figure of merit of X-ray detectors because high sensitivity allows the detection of low-dose X-rays, which decreases the patient dose [14]. For this reason, Wei and Huang [9] reported an MAPbBr_3_ single crystal (MSC) surface passivated by a UV–O_3_ method with a sensitivity of 80 μC Gy_air_^−1^ cm^−2^ in 2016. In 2019, Li [15] achieved a sensitivity of 467 μC Gy_air_^−1^ cm^−2^ by performing an additional MSC annealing process. Several studies were conducted to increase the detection sensitivity even further [16,17,18,19]; however, its value strongly depends on the detector thickness [8] and X-ray energy. Therefore, the same conditions should be applied to compare the sensitivities of various detectors. The carrier transport properties of perovskite detectors are determined by the utilized self-powered semiconductor materials. Surprisingly, the ferroelectricity of perovskites enables the separation of photogenerated carriers at each electrode, owing to the inner electric field produced by a strong polarization effect [20,21]. The diffusion length in perovskite single crystals exceeds 175 μm [11] and their lifetime is greater than those of other semiconductor devices [12,22]. Liu [23] and Cao [24] examined the feasibility of self-powered photodetectors with (EA)_2_(MA)_2_Pb_3_Br_10_ and P(VDF–TrFE)/perovskite bulk heterojunctions, respectively. Wang [25] detected UV light using a self-powered EA_4_Pb_3_Cl_10_ 2D hybrid perovskite, and self-powered X-ray detectors were investigated by other research groups [26,27]. The results of these studies revealed that perovskites could be potentially used as self-powered devices for detecting charge-collecting type radiation (such as α-radiation, β-radiation, X/γ-rays, and even high-energy photons) at a zero-bias voltage owing to their long diffusion length and lifetime. Meanwhile, such detectors require high-resistivity materials because a high dark current reduces the signal-to-noise ratio and decreases the measurement accuracy of the photocurrent generated by radiation [28,29]. The high resistivity increases the detector sensitivity by increasing the signal-to-noise ratio and reducing the dark current. To reduce the dark current, high-bandgap materials, dopants for growing intrinsic semiconductors, and passivants for Schottky surface heterojunctions have been developed previously [13,30,31]. However, it is also possible to reduce the leakage current and increase the material resistance by eliminating surface defects and rough regions through post-processing of the as-grown semiconductor crystals [32,33,34].

In this study, we enhanced the MSC sensitivity by processing the MSC surface and then confirmed the feasibility of employing self-powered MSCs for the detection of X-rays with high energies exceeding 1 MeV, which are commonly used in medical radiotherapy. For this purpose, we treated the as-grown MSCs with different mechanochemical processes to increase the detector sensitivity and then compared the electrical, structural, photo-responsive, carrier transport, and defect properties of the resulting materials. Additionally, an ohmic contact was formed using an Au electrode to improve the MSC sensitivity, and the responses of the fabricated MSC detectors were determined by conducting X-ray irradiation studies.

## 2. Materials and Methods

### 2.1. Growth of MSCs

The perovskite MSCs were grown using a modified ITC method [35]. First, MABr (1.4 M, 99.99%, Greatcell Solar Materials, Queanbeyan, NSW, Australia) and PbBr_2_ (98%, Sigma-Aldrich Inc., St. Louis, MO, USA) were weighed at the same molar ratio and dissolved in N,N-dimethylformamide (10 mL, Sigma-Aldrich Inc., St. Louis, MO, USA) to prepare an MSC precursor. The obtained mixture was stirred at 300 rpm at room temperature for 24 h to completely dissolve the precursor components. After that, it was filtered with a 0.2-μm poly(tetrafluoroethylene) filter followed by heating at a rate of 2 °C per day. The produced MSC was extracted when the crystal reached a desired size, and the remaining solution was repeatedly used for the subsequent growth processes. A seed was placed at the beaker bottom to grow a new MSC in the previously used solution because no differences were observed between the XRD patterns of the seedless and seed-grown MSCs.

### 2.2. Treatment of MSCs

Different mechanical treatments were performed on the as-grown MSCs. To eliminate a protruding part and flatten the rough MSC surface, all MSC specimens were mechanically lapped with 1200-grit SiC sandpaper followed by chemo-mechanical polishing with Al_2_O_3_ + ethanol, Al_2_O_3_ + ethylene glycol, or glycerin. The processed MSCs were classified into the lapped-only, Al_2_O_3_ + ethanol-polished, Al_2_O_3_ + ethylene glycol-polished, and glycerin-polished MSCs, followed by rinsing with hexane. The Ag and Au electrodes were deposited by applying Ag paste (Chemtronics, Sejong, Republic of Korea) and Au evaporation, respectively. The contact between the MSC and PCB board was established using the Ag paste and Au line (Nilaco Corporation, Tokyo, Japan) with a thickness of 50 μm.

### 2.3. Characterization of MSCs

The crystal structures of the fabricated MSC and its powder were determined by XRD using an Ultima/SmartLab diffractometer (Rigaku, Tokyo, Japan). The MSC energy band diagram was obtained by UPS and UV-visible spectroscopy using Thetaprobe (Thermo Fisher Scientific, Waltham, MA, USA) and Cary 5000 (Agilent Technologies, Santa Clara, CA, USA) instruments, respectively. The steady-state PL spectra were recorded by a MicroTime-200 (PicoQuant, Kekuléstr, Berlin, Germany) confocal microscope. The current–voltage characteristics were determined with a Keithley 237 source meter for the measurements of dark current and photocurrent using a radioisotope or fluorescence lamp. The hole mobility–lifetime products were calculated by fitting the photocurrent values measured using the Am-241 isotope with an initial radioactivity of 91.62 μCi with the modified Hecht’s equation.

### 2.4. Simulation and X-ray Evaluation of MSCs

All X-ray simulations were conducted using the Matlab R2020b software. The calculated current depending on the detector resistivity (Appendix A) was set to a random value within a previously established range of error. Mass attenuation coefficients (MACs) of the studied materials were retrieved from the NIST database [36] and used in other simulations based on the Beer–Lambert law. From the obtained MACs, a penetration or absorption rate was estimated at an MSC density set to 3.8 g/cm^3^ [37]. During X-ray evaluation, a Keithley 2636B source meter was used to measure the MSC photocurrents generated by the diagnostic medical and high-energy X-ray sources (GXR-S (DRGEM) and TrueBeam (Varian), respectively). To obtain the high-energy X-ray data, the TrueBeam linear accelerator installed at Samsung Changwon Hospital was used at a source-to-detector distance of 100 cm and field size of 10 × 10 cm^2^.

## 3. Results and Discussion

### 3.1. MSC Characterization

The inset of Figure 1a shows the surface of MSC grown by inverse temperature crystallization (ITC) and then mechanochemically polished with glycerin. The written letters “MAPbBr_3_” are clearly visible on its surface despite the high MSC thickness of 2.3 mm. The crystallinity and band structure of the produced MSC were determined by X-ray diffraction (XRD), ultraviolet photoemission spectroscopy (UPS), and UV-visible spectroscopy. Figure 1a,b shows the XRD patterns of the MSC and its powder diffraction pattern recorded at room temperature (25 °C), which are consistent with the results of previous studies [34,35]. Figure 1a displays the (100) and (200) peaks centered at 15.26° and 30.42°, respectively. Unlike the small (200) peak, the (100) peak mainly contributes to the crystal structure, indicating that the MSC has a homogeneous cubic crystal lattice [35]. Moreover, the (100)-to-(200) peak ratio is equal to approximately 20:1, and the rocking curve of the (100) peak (Appendix A) exhibits a full width at a half-maximum of 0.048°, corresponding to a high degree of crystallinity. The powder XRD spectrum contains the peaks positioned at 14.88°, 21.12°, 30.06°, 33.72°, 37.06°, 43.06°, 45.82°, and 48.46°, which correspond to the (100), (110), (200), (210), (211), (220), (300), and (310) planes, respectively. Thus, the XRD patterns of the MSC and its powder confirm the successful ITC growth of MAPbBr_3_.

To examine the bandgap characteristics of the MSC, its secondary electron cut-off energy (*E_cut-off_*) was measured by UPS (Figure 1c,d). The Tauc plot depicted in Figure 1e shows the MSC bandgap (*E_g_*). The Fermi level (*E_f_*) was calculated by subtracting *E_cut-off_* (16.3 eV) from the energy of incoming photons (21.22 eV). Valence band maximum (*E_VBM_*) and conduction band minimum (*E_CBM_*) were calculated considering both cut-off lines in Figure 1d,e. The estimated *E_VBM_*, *E_CBM_*, and *E_f_* values are equal to −5.73, −3.51, and −4.92 eV, respectively. Figure 1f displays the complete band diagram of the grown MSC, which exhibits p-type semiconducting properties. However, the as-grown MSC possesses a relatively rough surface; therefore, various mechanochemical surface treatments using different abrasives were performed as described in the Methods section.

### 3.2. Effect of Surface State on Radiation Detector Performance

Changes in surface state after the mechanochemical treatment of as-grown MSCs were evaluated by atomic force microscopy (AFM) studies, as shown in Figure 2. Figure 2a–c represents the 2D surface image of MSCs polished with Al_2_O_3_ + ethanol, Al_2_O_3_ + ethylene glycol, and glycerin, respectively, and Figure 2d–f shows each 3D image plot with the normalized *Z*-axis. As-grown MSCs could not be observed by AFM due to the giant gradients of surface roughness. The surface root-mean-square roughnesses (*R_q_*) of each surface were determined to be 58.36 nm, 106.03 nm, and 36.29 nm, respectively, which demonstrates a significant difference in the surface roughness depending on the type of polishing abrasives. In particular, the samples polished by either Al_2_O_3_ + ethanol or Al_2_O_3_ + ethylene glycol had an irregular surface with distinct valleys and hills (green dotted circles in Figure 2). The samples treated by glycerin alone showed a dense and more uniform surface with a *R_q_* of 36.29 nm. The drastic improvement of surface state by glycerin polishing can also be clearly verified from Figure 2g–i showing the roughness histograms (*y* axis) depending on the lateral distance (*x* axis) of samples. In addition, samples with low surface roughness minimize diffuse reflection, allowing visible photons to pass through the sample, making the sample more transparent (inset of Figure 1a). Additional atomic and chemical surface analysis of the glycerin-polished sample may be required, but for simplicity and clarity, the following section instead analyzes the effect of sample thickness on radiation detector performance in a more intuitive manner.

Figure 3 illustrates the effect of the surface state of MSCs on their electrical and transport properties, which are important for their potential applications as radiation detectors. Figure 3a,b shows the current distributions simulated at different detector resistivity (*ρ*) values (Appendix A). At each illumination point, the photocurrent (∆*I*) generated by radiation was set to exactly 2, as explained in Appendix A. However, the photocurrent of the low resistivity detector (∆*I*_2_) is lower than that of the high resistivity detector (∆*I*_1_) because the dark current (or leakage current) disturbs the measurements of the *I_on_* and *I_off_* terms in Appendix A. Moreover, the maximum error obtained for the low resistivity detector during photocurrent calculations can be twice as large as the set value of 5% because of the overlapping error bars of *I_on_* and *I_off_*. Thus, the high leakage current results in different photocurrent values measured under the same illumination conditions, deteriorating the precision and performance of the detector.

Another influence of the rough MSC surface on the performance of the radiation detector is the different transport properties of charge carriers in different regions. One of the main reasons for polishing the surface of a semiconductor single crystal is the homogeneous deposition of an electrode [32] along the surface shape. Thus, a rough surface produces an uneven interface after deposition (Figure 3c). When an external voltage (*V*) is applied to the electrode for the collection of photogenerated carriers, different electric fields are generated in the regions with different crystal thicknesses. For example, in Figure 3c, *e*_1_ and *e*_2_ are the photogenerated electrons that are fully identical except for their locations. These electrons are collected by the electric force produced by the electric field. However, they travel different distances (*d*_1_ ≠ *d*_2_), indicating that the electric fields in these two areas are also different (*E*_1_ ≠ *E*_2_) at the same bias voltage (*V* = *V*_1_ = *V*_2_) (Figure 3d). Furthermore, the path length (μτE) of electrons also depends on the carrier position, and the drift current density varies as *qE*(*nμ_e_* + *pμ_h_*) (Appendix A). Thus, the mechanical lapping and polishing processes, which smoothen the crystal surface, significantly affect the carrier transport and electrical properties of the material. The rough surface influences the performance of electronic devices, which collect carriers in the form of a drift current (such as semiconductor radiation detectors).

### 3.3. Enhancement of Defect, Transport, and Photocurrent Properties by Glycerin Polishing

Figure 4a shows the transmittance rates of MSCs subjected to different treatments as described in the Methods section. To normalize the intensity of incoming photons, the customized plates depicted in Appendix A were used in UV-visible spectroscopy measurements. The calculated Eg of the grown MSC is equal to 2.223 eV (Figure 1e); therefore, the photons with energies lower than Eg penetrate the MSC bulk without absorption. However, the measured transmittance rates strongly depend on the treatment process and polishing abrasive (Figure 4a). This effect is more clearly observed using the logarithm scale graph and photographs of the differently treated MSCs (Appendix A).

According to Ref. [38], the surface conditions of a single crystal influence the intensity of steady-state photoluminescence (PL), and the steady-state PL spectra obtained in this work (Figure 4b) show the same trend. The glycerin-polished MSC exhibits the highest PL intensity, while the MSCs treated by the other methods demonstrate lower PL intensities for the following reasons. First, the formation of excitons was suppressed by lapping the MSC surface or treating it with an Al_2_O_3_ + ethanol or Al_2_O_3_ + ethylene glycol mixture because the diffuse reflection from the MSC surface reduced the PL intensity. Only the glycerin-polished MSC demonstrated good hole transport properties (Figure 4c), which were evaluated using a modified Hecht’s equation [39,40]. Theoretically, the mobility–lifetime product of a hole (μτ_h_) is considered an invariant value that does not depend on the surface treatment process because it represents a bulk property [9]. However, the photogenerated carriers are trapped and recombined by surface defects and the interface between the electrode and material bulk. Thus, the calculation procedure involving Hecht’s equation results in surface-dependent μτ values because this equation is typically used to fit the collected current influenced by defects [40,41]. The μτ_h_ values of the other MSCs depicted in Appendix A are lower than that of the glycerin-polished MSC. Therefore, glycerin polishing improved not only the surface roughness but also the transport and defect properties of the produced MSCs.

To further examine the MSC surface processes, the current with/without visible light generated by a fluorescence lamp was measured for different MSC samples (Figure 4d and Appendix A). A photocurrent was successfully detected for all the polished MSCs regardless of the abrasive type (Figure 4d and Appendix A), whereas the lapped MSC produced a zero photocurrent (Appendix A). According to Figure 4a,b, the rough surface of the lapped MSC suppressed the absorption of visible light, which was supplied to each specimen by the fluorescence lamp. Figure 4d shows that the measured photocurrent (∆*I*) should become saturated at a certain bias voltage, while the photocurrent increase rate gradually decreases with increasing bias because the intensity of visible light produced by the fluorescence lamp is limited and constant. However, only the glycerin-polished MSC exhibits the above-mentioned tendency (Figure 4d). Thus, the fine polishing of MSCs with glycerin improves their roughness and photocurrent as well as the transport and defect properties.

### 3.4. X-ray Sensitivity Evaluation

Before X-ray evaluation, theoretical simulations were conducted for an X-ray spectrum with 70 kVp (Figure 5a,b), which was utilized in the real X-ray evaluation procedure. The maximum number of counts is observed at 35.5 keV owing to the beam hardening effect of an additional 3-mm Al filter, while the maximum number of counts in an X-ray spectrum is normally obtained at an energy equal to one third of the maximum energy in the medical X-ray range. Hence, the energy of 35.5 keV was selected for penetration rate simulations because it is predominant in the studied X-ray spectrum. Figure 5a shows the attenuation rate of the photons with an energy of 35.5 keV plotted as a function of the MSC thickness. The absorption intensity of the MSC with a thickness of 2 mm exceeds 97%. Figure 5b displays the mass attenuation coefficients obtained from the NIST database [36], input spectrum, and MSC absorption spectra recorded at various MSC thicknesses. Based on the simulation data, we employed a Ag/MSC/Ag device with an MSC thickness of 2 mm (Figure 5c) subjected to the lapping and glycerin polishing processes for X-ray evaluation. The perovskite exhibits outstanding charge carrier mobility regardless of its atomic composition [42]; however, the mobility of holes in the perovskite structure is better than the mobility of electrons [43]. The MSC band structure corresponds to that of a p-type semiconductor (Figure 1f) with holes serving as major carriers. To reduce the dark current acting as a noise in the radiation detector output [5,28,44], we designed a Schottky contact at the interface between the MSC bulk and Ag electrode, and the fabricated device was attached to a customized PCB board. The resistivity of the Ag/MSC/Ag device was 9.15 × 10^7^ Ω·cm, which corresponded to a current of 0.07 μA at a bias voltage of 5 V (Appendix A).

Figure 5d shows the X-ray responses of the Ag/MSC/Ag device obtained at various bias voltages ranging from 1 to 5 V. Measurable photocurrents were detected for all pulses. The effect of the bias voltage on charge collection is illustrated in Figure 5e,f. In Figure 5e, the intensity of the collected signal is weaker than the ideal signal, which originates from the charge loss due to the trapping and recombination of carriers at defects. Meanwhile, the collected signal stronger than the ideal signal is caused by the detrapping of charge carriers from the defects having a shallow trap level in the band diagram [45]. However, in Figure 5f, the collected signal is close to the ideal square pulse because the high bias voltage of 5 V increased the drift velocity (μE) of the collected carriers and reduced their defect exposure time. Therefore, the charge loss decreased with the reduced trapping of charge carriers followed by the decrease in the signal intensity due to detrapping. The rise and fall times of Ag/MSC/Ag measured at 2 V (Figure 5e) are 0.384574 and 0.551781 s, while the rise and fall times of Ag/MSC/Ag at 5 V (Figure 5f) are equal to 0.100325 and 0.033445 s, respectively. The generated photocurrent increases with an increase in the bias voltage (as shown in Figure 5g), but the R^2^ value of this curve amounts to 0.8818 reflecting its linearity. The calculated MSC sensitivity is 4.00 μC Gy_air_^−1^ cm^−2^ at 5 V (corresponding to an electric field strength of 25 V/cm), as shown in Figure 5h.

To improve the device sensitivity, the Ag electrodes were replaced with Au electrodes (Figure 6a). The Ag/MSC/Ag structure reduced the dark current by forming a Schottky contact at the interface between the MSC bulk and Ag electrode (Figure 6b). However, the modified structure can influence both the dark current and collection of the photogenerated charge carriers. The low surface roughness and dark current enhance the radiation detection performance (Figure 3); however, its prerequisite is a constant photocurrent (∆*I*). Therefore, we designed the Au/MSC/Au structure with an ohmic contact (Figure 6b) between the MSC bulk and electrode surface despite its larger dark current, which was compensated by a significant increase in the collected photocurrent (Figure 6c). Thus, the signal-to-noise ratio and collected photocurrent in the Au/MSC/Au structure were dramatically increased, as shown in Figure 5d,g and Figure 6d,e. The shape of the X-ray pulses obtained with the Au/MSC/Au device (Figure 6d) is close to that of the ideal square pulses generated previously (Figure 5d–f), despite the low applied voltage of 1 V. The linearity of the photocurrent dependence on the bias voltage is also better than that of the curve obtained for the Ag/MSC/Ag device, as indicated by the R^2^ value of 0.9969 (Figure 6e). Figure 6f shows the sensitivities of the Au/MSC/Au device, which reach 541 μC Gy_air_^−1^ cm^−2^ at a bias of 5 V (corresponding to an electric field strength of 25 V/cm).

### 3.5. Megaelectron Volt X-ray Detection at a Zero Bias

Additionally, we measured the signal generated by X-rays with an energy of 70 keV at a zero bias (Appendix A). Amazingly, the obtained current was direction-dependent, while the X-ray response under the applied bias exhibited only positive values (Figure 6d). After we changed the direction of the detection circuits, the current flowed in the opposite direction at a zero bias. Such a direction-dependent signal under a zero bias confirms the feasibility of the self-powered radiation detector because the internal electric field caused by perovskite polarization separates the generated carriers without an externally biased voltage [20,21]. In previous studies, self-powered devices containing perovskite materials were applied as photodetectors and visible-blind UV detectors [23,24,25]. Moreover, signals from γ-rays and X-rays were obtained using self-powered perovskites [26,27,46], which supports the feasibility of employing the self-powered radiation detector for detecting signals presented in Appendix A. In this section, we applied the MSC detector to X-rays with energies greater than 1 MeV.

To examine the attenuation properties of MSCs in the high-energy range, additional simulations were conducted (Figure 7a–c). The attenuation properties of MSCs allow accurate detection of high-energy X-rays, and the MSC with a thickness of 2 mm can absorb approximately 5% of the X-rays with an energy of 1 MeV (Figure 7b,c). Figure 7c shows the 6 MVp X-ray spectrum [47] generated by the high-energy X-ray generator described in Appendix A and absorption spectrum simulated for the 2-mm MSC detector. The absorbed fraction is smaller than that of the soft X-ray region used in medical diagnostic X-rays (Figure 5b) owing to the high transmission efficiency of hard X-rays. However, the attenuation of X-ray photons is accurately described by the multi-exponential function of the Beer–Lambert law, and the high-energy X-ray spectrum represents a continuous spectrum produced by both soft and hard X-rays [5,36]. Thus, the MSC detector can absorb X-ray photons and convert them to electrical signals to determine the original intensity or radiation dose. Appendix A shows the simulated 10 and 15 MVp X-ray spectra, respectively, which confirm the feasibility of detecting X-rays with energies higher than 1 MeV using the MSC detector.

The radiation produced by high-energy X-rays is different from that of soft X-rays. Because the secondary electrons generated by the primary incoming X-rays have a relatively high energy, they penetrate materials on the centimeter scale, although the secondary electron is the volumed particle [5,48]. Thus, the maximum dose (obtained from both primary X-rays and secondary electrons) is detected in the material bulk and not on the surface (Appendix A). To compensate for the effect of high-energy X-rays by adjusting the maximum dose position on the MSC detector, we installed solid water phantoms and a bolus with appropriate thicknesses, which are equivalent to the soft tissues in the human body (Figure 7d and Appendix A). The responses of the MSC device to high-energy X-rays (6 MVp) obtained at a zero bias voltage are shown in Figure 7e,f. In Figure 7e, the square pulse area of the photocurrent signal measured at the same dose rate of 400 MU/min is proportional to the absorbed dose. Figure 7f shows that the photocurrent generated at the same dose increases with increasing dose rate while maintaining similar square pulse areas (total dose). Appendix A displays the current signals obtained for the 10 and 15 MVp X-ray spectra using the treated MSC detector. It shows that the MSC X-ray detector can respond to X-rays with a maximum energy of 15 MeV.

## 4. Conclusions

In this study, we investigated the effect of mechanochemical surface treatment on the MSC detection performance and significantly increased the MSC sensitivity using the Au/MSC/Au configuration. The glycerin-polished MSC exhibited the highest PL intensity and transmittance rate of visible light as compared with those of the other treated MSC samples, which indicated high mobility-tau product, surface roughness, and higher photocurrent from visible light. The sensitivity of the fabricated MSC detector was higher than those of annealed MSCs reported in the literature, although the annealing step was not performed in this work. This means that surface and interfacial processes considerably affect the performance of radiation detectors, leaving room for further improvement of MSC properties by conducting additional annealing and/or interface passivation processes. More interestingly, both soft and hard X-rays were successfully detected at a zero-bias voltage. The continuous polarization induced by the internal electric field of MSCs promotes the movement of photocarriers to the opposite electrodes and their subsequent collection, leading to the generation of electrical signals. The direction-dependent photocurrent obtained in this work supports the above-mentioned carrier collection mechanism, which is also applicable to other perovskite single crystals with strong polarization-induced electric fields. Therefore, this work can serve as a guide for developing other self-powered perovskite high-energy X-ray detectors.

## Figures and Tables

**Figure 1 nanomaterials-13-02157-f001:**
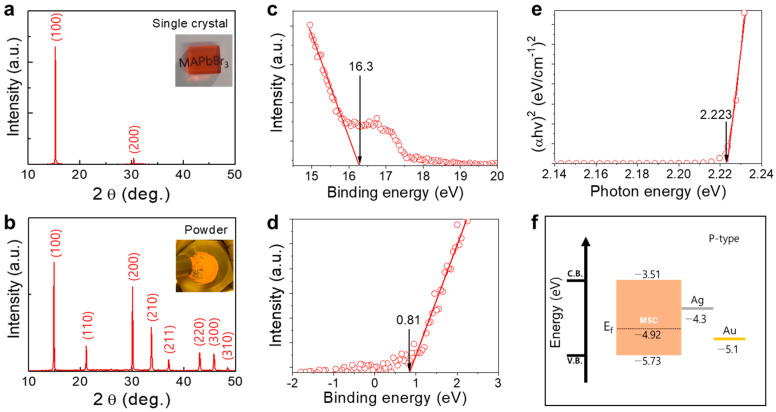
MSC characterization. XRD patterns of the (**a**) MSC and (**b**) its powder. UPS spectra showing the (**c**) *E_cut-off_* and (**d**) *E_f_* values. (**e**) Tauc plot constructed to calculate the *E_g_* value. (**f**) MSC band diagram obtained from the UPS data and Tauc plot.

**Figure 2 nanomaterials-13-02157-f002:**
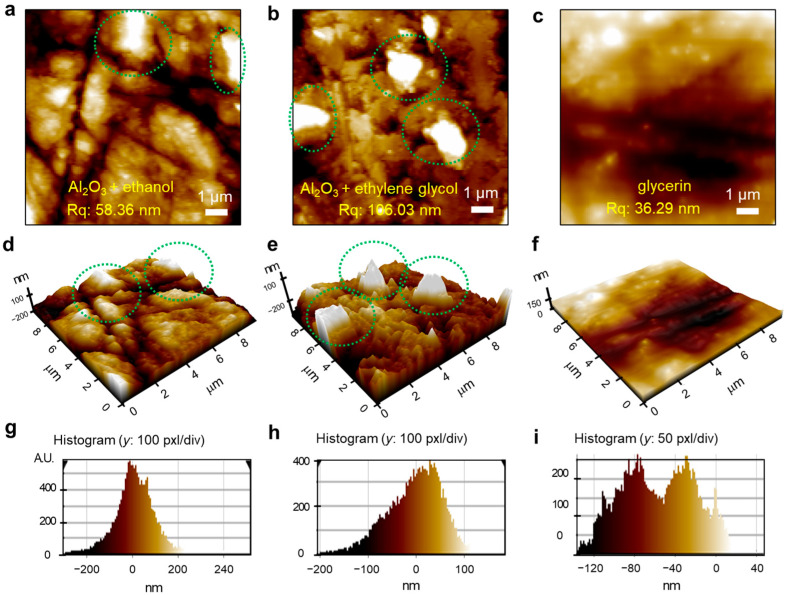
Atomic force microscopy measurements. (**a**–**c**) Two-dimensional surface images of MSCs polished by Al_2_O_3_ + ethanol, Al_2_O_3_ + ethylene glycol, and glycerin, respectively. (**d**–**f**) Three-dimensional image plots in order with normalized *Z*-axis. (**g**–**i**) Histograms of each MSC.

**Figure 3 nanomaterials-13-02157-f003:**
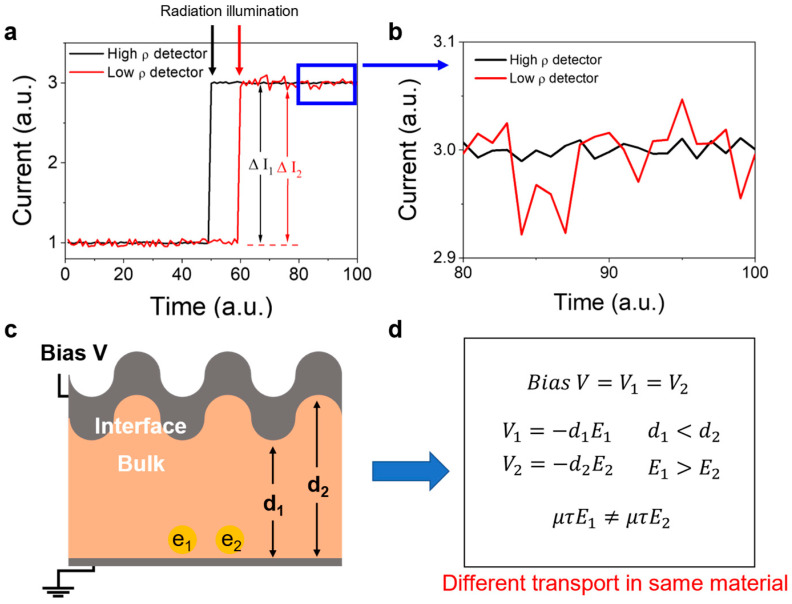
Surface roughness and its influence on the radiation detector performance. (**a**,**b**) Currents generated by the detectors with different resistivities under illumination. (**c**) Schematic of the rough surface and interface between the Ag electrode and bulk phase. (**d**) Dependence of the carrier transport properties on the carrier position.

**Figure 4 nanomaterials-13-02157-f004:**
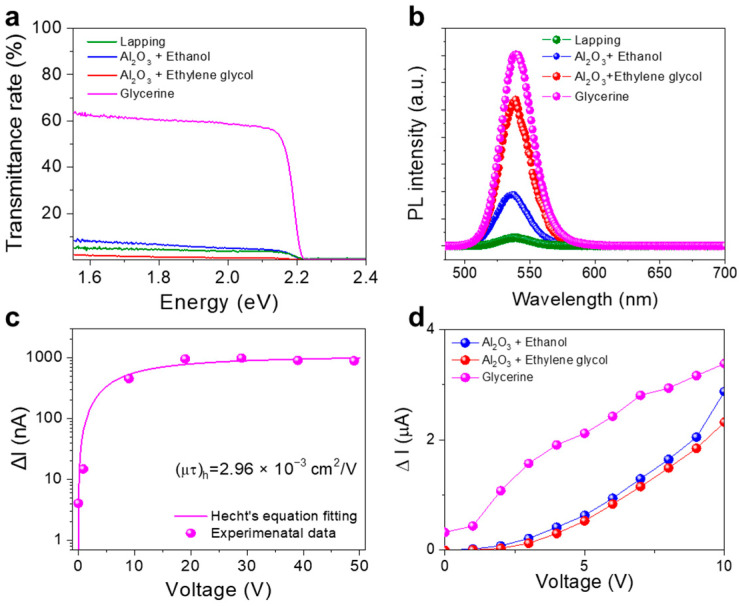
Dependence of the MSC properties on the polishing abrasive type. (**a**) Transmittance rates. (**b**) Steady-state PL spectra. (**c**) Photocurrents generated by the γ-rays emitted by the Am-241 isotope and Hecht’s equation fitting curve. (**d**) Photocurrents obtained from the fluorescence lamp depending on the surface treatment.

**Figure 5 nanomaterials-13-02157-f005:**
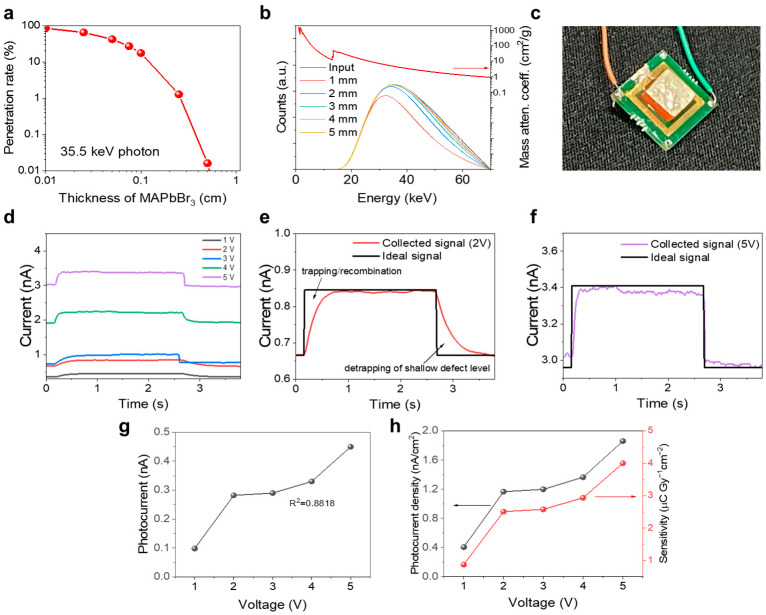
X-ray evaluation using the Ag/MSC/Ag device. (**a**,**b**) X-ray spectra and their attenuation simulated with the Matlab software. (**c**) Typical Ag/MSC/Ag device used for X-ray evaluation. (**d**–**f**) X-ray responses of the MSC device obtained by turning the X-ray source on and off. (**g**) Photocurrent and (**h**) sensitivity of the MSC device with the best performance plotted as functions of the bias voltage.

**Figure 6 nanomaterials-13-02157-f006:**
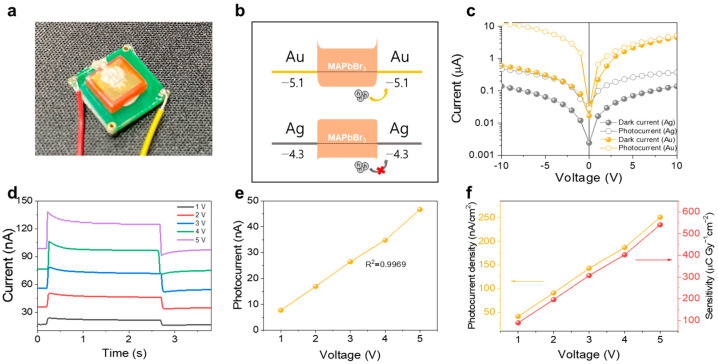
X-ray evaluation performed using the Au/MSC/Au device. (**a**) A typical Au/MSC/Au device used for X-ray evaluation. (**b**) MSC band structures and (**c**) I–V characteristics obtained for different electrodes. The MSCs had the same dimensions and were not identical to the samples used for X-ray evaluation. (**d**) X-ray responses of the MSC devices recorded by turning the X-ray source on and off. (**e**) Photocurrent and (**f**) sensitivity of the MSC with the best performance plotted as functions of the bias voltage.

**Figure 7 nanomaterials-13-02157-f007:**
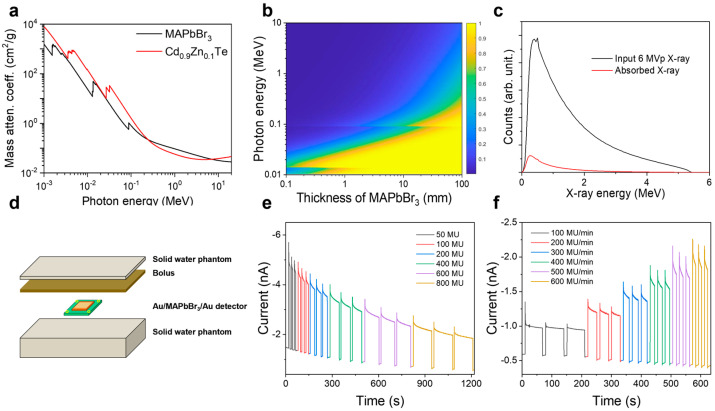
High-energy X-ray detection with the Au/MSC/Au detector. (**a**) Mass attenuation coefficients of the MSC and CdZnTe obtained in the high-energy X-ray region. (**b**) Simulated absorption rate as a function of the photon energy and MSC thickness. (**c**) Simulated spectra of the incoming X-rays [47] and X-rays absorbed by the 2-mm MSC. (**d**) Geometry of the high-energy X-ray detector. High-energy X-ray responses of the MSC plotted as functions of the (**e**) dose and (**f**) dose rate. MU is one of the dose units described in Appendix A.

## Data Availability

Data available on request from the corresponding author.

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
