# Peer review of "Solution-Grown MAPbBr3 Single Crystals for Self-Powered Detection of X-rays with High Energies above One Megaelectron Volt"

_nanomaterials, 2023, doi:10.3390/nano13152157_

Round 1
Reviewer 1 Report
In this work, the authors Park et al have prepared MAPbBr3 single crystals and characterize the corresponding detectors for X ray detection with various energy ranges. Currently the perovskite for hard X ray radiation detection have attracted enormous attentions due to their great potential. This work in general was systematic and interesting for researchers in the field of materials science and nuclear engineering. However, essential revisions are needed before publication in my opinion.
Here are the detailed comments:
Line 133, please clarify why (100) peak contributes to the crystal structure.
Line 145, please clarify the discrepancy between the value in Figure 1c and Evbm.
Figure 2c, the roughness after glycerin was improved however, the roughness with Rq of 36 nm is still a little high may need further refinement.
Figure 3c, 3d, the variation in carrier transport attributed to surface roughness might not be reasonable. In 3d, d1 is not equal to d2, however, the actual thickness is around 2 mm or even larger, the fluctuation in surface is on the order of hundreds of nm. So d1 is equal to d2 in practical. Please refine the discuss in above paragraph.
In the section device measurement under X rays, it is confusing to see the build-in field could be form. There is no difference in work function on two surfaces due to the use of same metals. Please clarify the results why the build-in field could be formed.
Author Response
Answers to reviewer (#1) questions
In this work, the authors Park et al have prepared MAPbBr3 single crystals and characterize the corresponding detectors for X-ray detection with various energy ranges. Currently the perovskite for hard X-ray radiation detection have attracted enormous attentions due to their great potential. This work in general was systematic and interesting for researchers in the field of materials science and nuclear engineering. However, essential revisions are needed before publication in my opinion.
Here are the detailed comments:
Line 133, please clarify why (100) peak contributes to the crystal structure.
Answer) Because the single crystallinity of present cubic MAPbBr3 single crystal (SC) is high, only two peaks, (100) and (200), were identified in the XRD pattern of SC. This means the SC orientation along the 100 direction of cubic crystal lattice. Furthermore, a high peak ratio, (100)/(200), means an excellent single crystallinity.
Line 145, please clarify the discrepancy between the value in Figure 1c and Evbm.
Answer) For clarifying, the following is added on line 145, which is shown as red letters. In addition, the order in Figure 1 was changed (Figure 1c and d). Please check our revised manuscript.
“The Fermi level (Ef) was calculated by subtracting Ecut-off (16.3 eV) from energy of incoming photons (21.22 eV). Valence band maximum (EVBM) and conduction band minimum (ECBM) were calculated considering both cut-off lines in Figure 1d and e.”
Figure 2c, the roughness after glycerin was improved however, the roughness with Rq of 36 nm is still a little high may need further refinement.
Answer) Thank you for good comment. We are going to use a more precise lapping machine. But, in this manuscript, we compared the surfaces with different abrasives and the Rq of 36 nm is the minimum roughness.
Figure 3c, 3d, the variation in carrier transport attributed to surface roughness might not be reasonable. In 3d, d1 is not equal to d2, however, the actual thickness is around 2 mm or even larger, the fluctuation in surface is on the order of hundreds of nm. So d1 is equal to d2 in practical. Please refine the discuss in above paragraph.
Answer) Thank you for good suggestion. If the SC of 2 mm thick is a perfect rectangular shape, the hundred nm scale difference can be neglected. However, the interpretation of Fig 3 has a role to explain the roughness of crystal and its effects, because it is not that as-grown single crystal always have ideal cubic shape to apply on X-ray detector, as shown in Figure R1. Thus, Figure 3 in manuscript can serve explanations not only for differently polished crystal in this manuscript but also all the cases with typical polishing.
Figure R1. Photographs of typical as-grown MAPbBr3 single crystal (a, b) each side of crystal, (c) Cross section of crystal with Green line in (b).
In the section device measurement under X rays, it is confusing to see the build-in field could be form. There is no difference in work function on two surfaces due to the use of same metals. Please clarify the results why the build-in field could be formed.
Answer) According to the reference [23] in this manuscript, continuous polarization can cause an inner electric field, which drive charges carriers to each electrode. The meaning of “built-in field” may be confusing to the reader, so we changed the built-in field into “polarization-induced internal electric field”. Please see lines 18 and 387.
Please see the attached response file

Reviewer 2 Report
The authors reported perovskite single crystals for X-ray detection application. They demonstrated a mechanochemical surface treatment approach using glycerin as well as improved electrode design for improving the performance of MAPbBr3 single crystal-based X-ray detector, particularly in charge carrier diffusion length, photocurrent and interfacial electrical contact. A bias-free X-ray detector was also demonstrated under X-rays of over 1MeV, indicating the potential of the perovskite single crystals for self-powered X-ray detector application. I would like to recommend to publish this work by addressing the following comments.
1. Is there any post-cleaning process after the chemo-mechanical polishing step before the Au/Ag evaporation?
2. On page 6, line 224, the authors mentioned that “….the diffuse reflection from the MSC surface reduced the PL intensity.” This claim is not convincing enough simply from the roughness factor of the crystal surface, because the contribution of the diffuse reflection to the overall emitted PL signal should be estimated for achieving this conclusion.
3. How`s the stability of the perovskite single crystal on exposure to these three different solvents? XPS and XRD are recommended to compare the surface chemistry and bulk crystal phase of the single crystals before and after the chemo-mechanical polishing treatment.
4. On Page 10, line 372, “…which indicated good defect, carrier transport, and photocurrent properties.” This is a bit confusing for general readers. It`s better to state clearly what`s the good properties in terms of defect, carrier transport and photocurrent? For example, lower defect density (surface or bulk?), higher photocurrent, longer carrier diffusion length? Also, Time-resolved PL is highly recommended to support this conclusion.
5. How`s the device long-term operational stability or storage stability?
Author Response
Answers to reviewer (#2) questions
The authors reported perovskite single crystals for X-ray detection application. They demonstrated a mechanochemical surface treatment approach using glycerin as well as improved electrode design for improving the performance of MAPbBr3 single crystal-based X-ray detector, particularly in charge carrier diffusion length, photocurrent and interfacial electrical contact. A bias-free X-ray detector was also demonstrated under X-rays of over 1MeV, indicating the potential of the perovskite single crystals for self-powered X-ray detector application. I would like to recommend to publish this work by addressing the following comments.
Is there any post-cleaning process after the chemo-mechanical polishing step before the Au/Ag evaporation?
Answer) Thank you for a good question. In all process, the cleaning with Hexane was carried out. We added in line 95 (Materials and Method section) the following sentence: “, followed by rinsing with haxane”. Please see line 95.
On page 6, line 224, the authors mentioned that “….the diffuse reflection from the MSC surface reduced the PL intensity.” This claim is not convincing enough simply from the roughness factor of the crystal surface, because the contribution of the diffuse reflection to the overall emitted PL signal should be estimated for achieving this conclusion.
Answer) As shown in Supporting Information Figure S2c, (only-lapped / Al2O3+ethanol / Al2O3+Ethylene glycol) MAPbBr3 single crystal show the nontransparent surface, which means the visible lights don’t transmit. With this evidence, the diffuse reflection was proposed one of the possible reasons to suppress the PL intensity; decrease of the absorbed photons reduces the generation of exciton, thereby causing lower PL intensity.
How`s the stability of the perovskite single crystal on exposure to these three different solvents? XPS and XRD are recommended to compare the surface chemistry and bulk crystal phase of the single crystals before and after the chemo-mechanical polishing treatment.
Answer) We appreciate a nice comment. Unfortunately, the effect of different solvents was not discussed further in this manuscript. Only glycerin-polishing was discussed because it produced the best surface condition and properties. This comment can be performed in next study.
On Page 10, line 372, “…which indicated good defect, carrier transport, and photocurrent properties.” This is a bit confusing for general readers. It`s better to state clearly what`s the good properties in terms of defect, carrier transport and photocurrent? For example, lower defect density (surface or bulk?), higher photocurrent, longer carrier diffusion length? Also, Time-resolved PL is highly recommended to support this conclusion.
Answer) We changed the sentence to “high mobility-tau product, surface roughness, and higher photocurrent from visible light” to avoid redear confusion. In our next report, we will consider time-resolved PL study.
How`s the device long-term operational stability or storage stability?
Answer) It is well-known that the MAPbBr3 perovskite single crystals have less stability issues compared with iodine-based perovskite single crystals (i.e., FAPbI3); thus, we did not verify the quantitative stability of the MAPbBr3 single crystals. However, they were still functional after 2 months and showed similar magnitudes of photocurrent, although this is not included in the manuscript.

Round 2
Reviewer 1 Report
The authors have made reasonable response to the comments. And I recommend the following publications as it is.